

# Greenland's annual and interannual mass variations from GRACE/GRACE-FO linked with climatic indices

Florent Cambier[1], José Darrozes[1], Muriel Llubes[1], Lucia Seoane[1], Guillaume Ramillien[1]

[1]Université Toulouse III - Paul Sabatier, Observatoire Midi-Pyrénées, GET, UM5563, CNES/CNRS/ID, 14 Avenue Edouard Belin, 31400 Toulouse, France

*Correspondence to*: Florent Cambier (florent.cambier@get.omp.eu)

**Abstract.**

The ongoing global warming threatens the Greenland Ice Sheet (GIS). The GIS exhibits an overall mass loss since 1990. This ice mass loss varies annually and interannually, reflecting the intricate interactions between the ice sheet and the atmospheric and oceanic circulations. In this study, we look at the temporal variations of the GIS mass balance, from April 2002 to the end of 2023, using data from the Gravity Recovery and Climate Experiment (GRACE) and its Follow-On (GRACE-FO) missions. We analyze the common cycles and the connections between GIS mass changes, climatic indices, and meteorological parameters, namely: North Atlantic Oscillation (NAO), Greenland Blocking Index (GBI), Atlantic Multidecadal Oscillation (AMO), temperature, precipitation, and surface albedo. Each variable is cumulated over time to align with the monthly mass variations since 2002. By applying Empirical Orthogonal Functions to mass variation data derived from the International Combination Service for Time-variable Gravity Fields (COST-G) solution, we identified five principal modes of variability, explaining 67.5% of the total variance. The primary mode captures annual and interannual frequencies, ranging from 4 to 11 years, while subsequent modes only add information on the interannual part. Wavelet Analysis reveals significant annual correlations between ice mass changes and temperature (r = -0.88), NAO (r = 0.74) or GBI (r = -0.85). There are also notable lags such as a 3.5 year delayed response of the AMO to GIS mass variations during the 22 years of data. The delayed response can be linked to the time needed for Greenland's lost mass of low-density water to reach the area where the AMO index is calculated. We also suggest an annual cycle connecting the GIS mass changes, the climatic indices, and the meteorological parameters. On the other hand, we observe complex interannual variations, including the role of sea surface temperature and atmospheric pressure in modulating ice mass balance, temperature, and precipitation. Results also indicate that cycles of 11 years in NAO, GBI, and temperature are strongly linked to solar activity. Furthermore, we observe cycles, lasting between 4 and 7 years, that align with studies linking them to atmospheric oscillations or the effect of the solid Earth's internal geodynamics.

## 1 Introduction

The Greenland Ice Sheet (GIS) is the greatest contributor to the current sea level rise after the thermic expansion of oceans (Cazenave et al., 2018). The rate at which the GIS is losing mass can largely be explained using a simple linear trend, as presented to the general public on the NASA website, showing a loss rate of -269 Gt/yr between 2002 and 2023. Seasonal, annual, and interannual variations are observable (Bevis et al., 2019; Bian et al., 2019; Forsberg et al., 2017; Otosaka et al., 2022; Simonsen et al., 2021; Wang et al., 2023; Wouters et al., 2008). Those variations are complex intricate responses (Wunderling et al., 2024) to internal and external changes in the environment of the GIS and its surrounding climatological systems, such as the surface temperature, the type and amount of precipitations, and the oceanic temperature (Ryan et al., 2023; Schmid et al., 2023). In 2018, an event of destabilization of the K.J.V. Steenstrups Nordre Bræ glacier was shown to be caused by an increase in oceanic temperature (Chudley et al., 2023). The influence of climate indices such as the Greenland Blocking Index (GBI), the North Atlantic Oscillation (NAO), and the Atlantic Multidecadal Oscillation (AMO), have been investigated.



For example, NAO is not the best predictor of precipitation in the South-East of Greenland (Berdahl et al., 2018) and it is evoked that both NAO and GBI are correlated to the mass loss (Välisuo et al., 2018).

Herein, we focus on the period between April 2002 and the end of 2023 with mass loss recovered from the monthly gravimetric satellite dataset of the Gravity Recovery and Climate Experiment (GRACE) and its Follow-On (GRACE-FO) mission. Those two missions use a pair of polar-orbiting satellites linked by microwave-ranging instruments. They monitor Earth's gravity field and grant us access to the monthly temporal variations of that field. At this scale, the changes in gravity are predominantly caused by hydrological displacement (Frappart et al., 2019; Ramillien et al., 2021; Seoane et al., 2013). GRACE and GRACE-FO data allow us to evaluate the mass balance of glaciers, ice caps, and ice sheets around the globe, particularly in our region of interest: Greenland (Bevis et al., 2019; Bian et al., 2019; Forsberg et al., 2017; Graf and Pail, 2023; Otosaka et al., 2022; Velicogna et al., 2020; Wang et al., 2023).

Our objective is to better our understanding of the dynamics linking indices (NAO, GBI, and AMO), and parameters of temperature, precipitation, and albedo, with annual and decadal mass variation of the GIS. With that aim, we implement a method in which an index is cumulated over time and compared to the modes of the Empirical Orthogonal Function (EOF) of the time series of ice loss (King et al., 2023). After presenting the different data, their treatment, and the EOF and cross-wavelet analysis method, we show the principal modes of variations and observe the most significant links between the mass changes and the climatic indices and meteorological parameters. In the discussion we explore in more detail the common cycles and investigate the behavior from those intertwined elements to each other, revealing a complete annual cycle.

## 2 Materials

### 2.1 GRACE/GRACE-FO

We chose the GRACE and GRACE-FO solutions, release one and two, respectively, from the International Combination Service for Time-variable Gravity Fields (COST-G), which combine those of other centers with different weights depending on the month. The data extend from April 2002 to December 2023 and is in the form of monthly spherical harmonics (SH) coefficients with a maximum degree of 90. They're available at the International Centre for Global Earth Models (ICGEM). (Meyer et al., 2019, 2020, 2023)

Those SH were processed to obtain a 1° grid over Greenland. The C20 coefficient is replaced by Satellite Laser Ranging (SLR) derived values and the geocenter coefficients, C10, C11, and S11 are inserted following the corresponding Technical Notes (Sun et al., 2016; Swenson et al., 2008). When recomposing the SH a Decorrelation and Denoising Kernel filter of level 5 (DDK-5) was applied to remove the stripping effect otherwise present (Kusche et al., 2009). This filter offers a good balance between smoothing and denoising, its effective diameter in Greenland's latitude is ~360 km, which is in the same range as the spatial resolution of GRACE.

We also need to correct the effect of the Glacial Isostatic Adjustment (GIA). We chose the most recent model from the ICE-xG series, named ICE-7G_NA (VM7), an upgrade from the precedent ICE-6G (Peltier, 2020; Roy and Peltier, 2018).

We use forward modeling to reduce the methodological leakage errors resulting from the recomposition of SH, which manifests as land-ocean leakage along the coast. This method was shown to be an effective solution to this problem over Greenland or even basin-wide areas. (Chen et al., 2015; Jin and Zou, 2015; Zou et al., 2020)

Temporal mass variations recovered from the GRACE COST-G solutions are sampled at a constant monthly timestep. However, there are temporal gaps in the time series, for the small ones, up to 2 months, the gaps are linearly interpolated.



Otherwise, for the large gap between GRACE and GRACE-FO, we use a frequency interpolation method commonly used for quasi-periodic signals. We fill this gap with the inverse Fast Fourier Transform (FFT) of the FFT coefficients estimated for this period. Those coefficients are obtained from the linear interpolation of the FFT coefficients of the preceding and following years of GRACE and GRACE-FO respectively. The completed Greenland mass change is presented in Fig. 1.

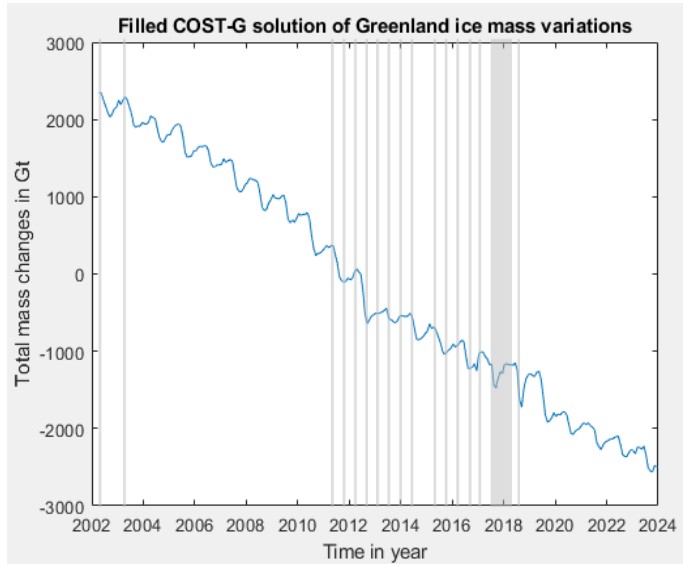

**Figure 1: Gap-filled ice mass variations over Greenland, from the GRACE COST-G solutions, expressed in Gt, from 2002 to 2024.**

**2.2 Climate Indices**

Over Greenland, three indices can directly participate in ice mass variations: NAO, GBI, and AMO. These indices are treated following a cumulative method used to interpret Antarctica's interannual ice mass variations (King et al., 2023). After a normalization, we temporally sum each index. We only considered the GRACE and GRACE-FO period data, so we truncated the portion before April 2002 or after December 2023. Then, we renormalize before finally removing a linear trend. The result is related to the cumulative mass variations observed by gravimetric satellites rather than the mass flux. Studies on ice shelf elevation changes, time-integrated surface mass balance, and Antarctic ice mass variations have already shown this. (Kim et al., 2020; King et al., 2023; Paolo et al., 2018)

**2.2.1 NAO**

The NAO is provided by the Climate Prediction Center from the National Center for Environmental Prediction of the National Oceanic and Atmospheric Administration (NOAA) and the National Weather Service. The dataset is the monthly mean value of NAO from January 1950 to the present. The NAO is related to the changes in the Azores High position. It's an index based on the difference in sea-level pressure between the Subpolar Low and the Subtropical High. This dipole is stronger in a positive phase and is weakened in a negative one. This index is calculated from the projection of the NAO loading pattern, which is




the first mode of a rotated EOF over 0° to 90° of latitude for the monthly mean anomaly data between 1950 and 2000, over the daily anomaly for the 500 mbar height field. (Barnston and Livezey, 1987; Chen and Dool, 2003; Dool et al., 2000)

The NAO phenomenon is known to be more negative in summer than in winter. It's correlated to Greenland's ice loss thanks to its connection to the temperature, meltwater runoff, and precipitations. Cycles of 7, 13, 20, 26 and 34 years are also present.
(Brils et al., 2023; Seip et al., 2019; Shang et al., 2022; Zhang et al., 2023)

### 2.2.2 GBI

The GBI is available through the NOAA's Physical Sciences Laboratory (PSL). The format is a daily coverage since 1948 from the NCEP/NCAR Reanalysis (Kalnay et al., 1996), which we then transformed into monthly values. This index concerns
the region over Greenland from 20 to 80° W and 60 to 80° N. It represents the mean 500 hPa geopotential height over Greenland which tells us if high-pressure are more or less blocked over this area. (Hanna et al., 2016)

Positive GBI values indicate an anticyclonic regime where the cloud cover is weakened and the Near-Surface Temperature (NST) rises (Hermann et al., 2020; Rowley et al., 2020). This index has been more positive since the '90s, it shows an interannual cycle length of 8 years and is correlated to temperature. (Hanna et al., 2016, 2021; Maure et al., 2023)


### 2.2.3 AMO

The AMO index comes from the PSL of NOAA. The monthly and unsmooth dataset spans from January 1856 to January 2023. It's a weighted average estimated over the North Atlantic Sea Surface Temperature (SST) using the Kaplan SST dataset (5x5). It encompasses the area between 0 and 70° N. (Enfield et al., 2001; Kaplan et al., 1998)

When this index rises, the temperature of the surface waters of the North Atlantic Ocean does the same. The rising sea water temperature shows an effect on the temperature and on the ice mass variation over Greenland. Atlantic water intrusions can also greatly influence glacier outlets, enhancing ice loss volumes by directly destabilizing glaciers with the contact of warmer and saltier waters. (Chudley et al., 2023; Li et al., 2022b).

**2.3 Meteorological Parameters**

To further our understanding of the dynamics between the GIS and the indices, we investigate meteorological parameters such as the NST, the volume of precipitations, and the surface albedo over the area of Greenland. Those three parameters are from the C3S Arctic Regional Reanalysis (CARRA) on the Climate Data Store of Copernicus Climate Change Service. (Schyberg et al., 2020)

All data from CARRA are at a 2.5 km horizontal resolution. We recovered 1 data point per day between the $1^{st}$ of January 2002 and the $31^{st}$ of December 2023. The three parameters we focus on are the 2 m temperature, the albedo, and the total precipitation. Those datasets were processed to a 1° grid and monthly temporal variations over Greenland.

Temperatures are used to evaluate the impact of heat on the GIS. We compute a time-weighted thermal availability (Eq. 1), a parameter that emphasizes the duration and intensity of positive temperatures.



$$TDT_{+°_C}(i) = TD_{+°_C}(i) \times \overline{T_{+°_C}(i)} \tag{1}$$

Where *TDT*, in °C, represents the available thermal energy to influence ice melting for the month (*i*) and only for positive temperature ($_{+°C}$). *TD* is estimated as the percentage of days per month (*i*) where positive temperatures ($_{+°C}$) are reached. It's multiplied by the average positive Temperatures ($T_{+°C}$) during the month (*i*).

The Albedo (Al) is the average of all surfaces' percentage of reflectance to solar radiation. It fluctuates with the freshness and

impurities in snow and ice or the physical states of water.

Finally, precipitations (P) are an important parameter to look at, because they are the only way to increase the ice mass on the GIS. They are issued from CARRA's forecasts. We use the accumulated mass per unit of the surface of all the different types of precipitation between each timestep.

**3 Methods**

**3.1 Empirical Orthogonal Function**

We use the EOF method to analyze the temporal ice mass variations. The data is decomposed into *N* orthogonal basis functions to account for as much variance as possible. They're found by computing the eigenvectors of the covariance matrix $Z^T Z$, in which Z represents the matrix temporal variations at each grid point, or by doing a singular value decomposition (Eq. 2). The

spatial pattern, the time series, and the eigenvalue which gives the percentage of explained variance are available with this method. (Ghil et al., 2002; Preisendorfer and Mobley, 1988).

$$Z(x,y,t) = \sum_{k=1}^{N} S_k(x,y) \cdot PC_k(t) \tag{2}$$

Where, *Z(x,y,t)* is ice mass variation in space *(x,y)* and time *t*; $S_k(x,y)$ is the eigenvector of mode *k* in the space *(x, y)*. It shows the spatialized structures that can account for the temporal variations of *Z*; $PC_k(t)$ is the principal component of the mode k

and entails how the amplitude of each mode varies with time.

This method is a consistent way to investigate the dominant modes of variability of interannual ice mass redistribution (King et al., 2023; Li et al., 2022a; Mémin et al., 2015). We removed a linear trend at each grid point *(x,y)* of the mass change obtained through SH, to focus on interannual variability. Then, the dataset is decomposed into temporal and spatial orthogonal oscillations, and the total variance explained by each EOF mode is estimated. We look for the inflection point of the eigenvalue

curve and the cumulative percentage of explained variance to evaluate how many principal components are significant.

**3.2 Cross Wavelet Transform**

We use a time-frequency representation to analyze and cross-correlate the temporal mass variations and the climatic/meteorologic variables defined in 2.2 and 2.3. We perform the Cross Wavelet Transform (XWT) analysis from the

"Cross Wavelet and Wavelet Coherence toolbox" (Grinsted et al., 2004). Its utility is to unveil common frequencies, the type of correlation, and the lead-lag (phase shift) between two temporal signals. Common frequencies are only shown if the two signals are correlated with a two sigma threshold.



Wavelet Transform (WT) analysis uses a mother function, $\psi$, which is then applied, with an increasing wavelength, like a band-pass filter to the time series. When the wavelet daughter and the signal correspond at a defined time, a peak appears on the analysis for the specified frequency.

We chose $\psi$ as a Morlet wavelet (Eq. 3) for its balance between time and frequency localization which is good for studying non-stationary physical signals like mass variations (Goupillaud et al., 1984; Grinsted et al., 2004; Lopez et al., 2018; Morlet et al., 1982; Xu and Burton, 2021).

$$\psi(\eta) = \pi^{-\frac{1}{4}} e^{i\beta\eta} e^{-\frac{\eta^2}{2}} \tag{3}$$

Where $\eta$ is the dimensionless time parameter defined as $\eta = n/s$, $n$ being the time and $s$ the wavelet's scale; $\beta$ is the dimensionless frequency set at 6 for a good balance between time and frequency localization (Torrence and Compo, 1998).

The XWT compares two WT of the time series $x$ and $y$ and is defined as the following Eq. (4):

$$W_n^{X\,Y}(s) = W_n^X(s)\,W_n^{Y*}(s) \tag{4}$$

Where $|W_n^{X\,Y}(s)|$ is the cross wavelet power at a time $n$ and for a wavelet scale of $s$. The complex argument $W_n^X(s)$ is the local phase of the signal $x$ and the * signifies the complex conjugate.

Assuming that our processed signals are cyclic over time. We can increase the precision of the frequencies and overcome the edge effect related to the wavelet's length, by duplicating the signals on both ends.

## 4 Results

When comparing the detrended GRACE mass loss to the climatic indices or the meteorological parameters, we can easily find a link at an annual frequency. Thus, decomposing the signal into dominant modes of variability is necessary to figure out interannual phenomena influencing the mass variations.

Here, we present the relations between the GRACE signal, the climate indices, and the meteorological parameters obtained from the study of EOF decomposition and XWT analysis. We recall that all those indices and parameters are cumulated.

### 4.1 GRACE EOF modes

Figure 2a to 2e shows the decomposition of the GRACE temporal signal over Greenland, in Gt, for the first five modes (M1, M2, M3, M4, and M5). The explained variance of those five is 31.9%, 17%, 8.5%, 5.7%, and 4.4% respectively, for a total of 67.5%. The latter modes are not studied further here, each expressing less than 4% of the variance. (Figure 2f).

In M1, we see annual oscillations, superimposed on a longer period variation. The annual oscillation amplitude is 20 Gt, whereas the amplitude is 50 Gt for the interannual one. The separation of the yearly component may not be perfect, and it could also be found in the other modes. But for those, higher frequencies are dominating and there is no clear annual signal anymore. (Figure 2a)

The pattern of M2 shows an interesting long-period variation, very different from that of M1. A different climate index intervention could explain this type of variation. Except for the high-frequency content, nothing else is visible on M3 and M4 plots. The last mode, M5, is different from the other four. It looks like a residual noise curve, except for two peaks in 2017 and



2018. These peaks reach 50 and 20 Gt respectively which is higher than the rest of the curve values. Mode 5 will be covered separately in the discussion section. Subsequently, we focus on the first four modes. (Figure 2)

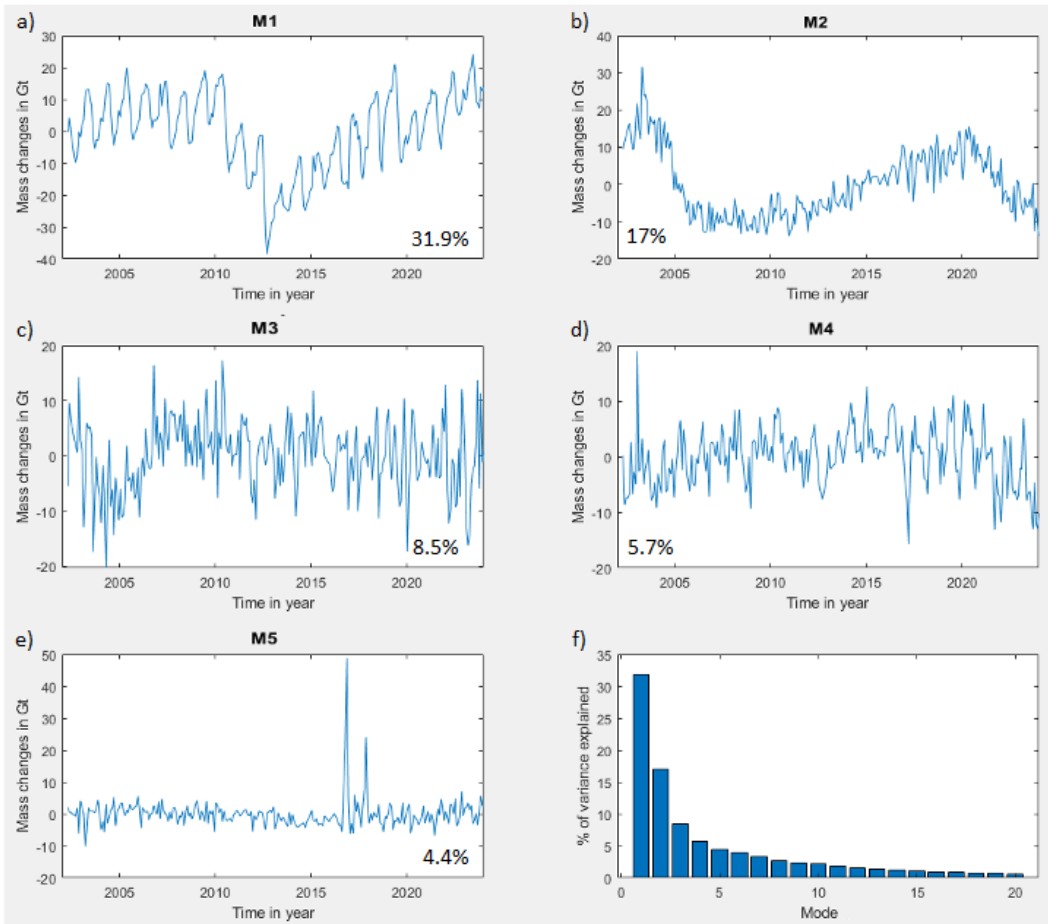

**Figure 2: Principal components of the EOF analysis of Greenland ice mass changes between 2002 and 2024. From a) to e) are the first five modes in order of explained variance and f) the bar representation of the explained variance of the first twenty modes.**

### 4.2 Links between ice mass changes and climate indices

Each climatic index has been compared to the EOF decomposition of the GRACE mass signal. This has been done for all the modes and all combinations of modes from M1 to M5. The relevant common periods and phase shifts are identified by XWT analysis. In addition, the correlation coefficient is calculated for these common periods. The most significant ones are given in Table 1.



| Common Frequencies | NAO | GBI | AMO | TDT | P | Al |
|---|---|---|---|---|---|---|
| **1 yr** | 0.74 | -0.85 | -0.61 | -0.88 | 0.54 | 0.9 |
| | *M1* | *M1* | *M1* | *M1* | *M1* | *M1* |
| **4-7 yr** | — | — | -0.77 | — | — | 0.8 |
| | | | *M1 to M4* | | | *M1 to M3* |
| **11 yr** | 0.77 | — | — | -0.71 | 0.75 | — |
| | *M1 to M4* | | | *M1 + M3 + M4* | *M1 to M5* | |
| **>15 yr** | — | -0.91 | 0.73 | — | — | — |
| | | *M1 + M2* | *M1 + M2* | | | |

**Table 1: Coefficients of correlated common frequencies (more than two sigma of confidence) between the GIS modes or sum of modes (M1, M2, M3, M4, and M5) and the climatic indices (North Atlantic Oscillation (NAO), Greenland Blocking Index (GBI), and Atlantic Multidecadal Oscillation (AMO)) or the meteorological parameters (time-weighted thermal availability (TDT), Precipitation (P), and Albedo (Al)). "to" means the sum of the encompassed modes. The highly positively correlated coefficient, superior to 0.7, and the highly negatively correlated ones, inferior to -0.7, are in red and blue respectively. Values between -0.5 and 0.5 are not represented.**

We plot the decomposed GRACE variations and one of the indices on the same figure in Fig. 3, Each time, the mode that presents the best overall correlation is chosen. This is M1 for NAO (a), and the sum of M1 and M2 for GBI (b) and for AMO (c). In addition, we represent the corresponding XWT to the right of each plot, the bold line shows the area of significant correlation with a confidence of two sigma.

The NAO has two common periodicities with M1. They're positively correlated for both but the link between them is not as strongly defined for the annual period, even if the corresponding correlation coefficient is 0.74. The second common period, around 11 years, shows a small phase lag of 1 year, which disappears when the XWT is performed between the NAO and the sum of M1 to M4. In this case, the correlation coefficient is 0.77. (Figure 3a and Table 1)

In Fig. 3b, the GBI has two common frequencies with M1 plus M2. It's worth noticing that both the annual and longer-term are strongly anti-correlated. The analysis indicates that the GBI leads by 1 or 2 months the mass variations at the annual scale (r = -0.85). For the interannual correlation, the XWT shows a significant link between the GBI and the GIS with a frequency of at least 15 years, it corresponds to the observable similarity between both signals over the 22 years of data. The corresponding correlation coefficient is -0.91. (Figure 3b and Table 1)

The AMO shows three common periods with the sum of M1 and M2. The annual frequency is less defined than that of the GBI but more than that of the NAO. The first interannual common period is for a 7 year periodicity. It's only significantly correlated until 2014 when comparing the AMO to the sum of M1 and M2. For this frequency, the period between 2014 and 2024 can be found by adding M3 and M4, but this decreases the correlation on the annual frequency. The coefficient of anti-correlation is -0.77 for this 7 year oscillation. The XWT shows there is another significant link at a frequency of at least 15 years like what is found for GBI. This one is positively correlated (0.73) and has a striking lag of 3.5 years on the AMO, this delay is observable on the plot showing both signals. (Figure 3c and Table 1)





**Figure 3: Comparison of the cumulated indices (in red) with the modes or summation of modes of ice mass changes in Gt (in blue) between 2002 and 2024, and, at the right, the corresponding XWT analysis. In a) is the NAO compared with the M1, b) is the GBI with M1 + M2, and c) is the AMO with the sum of M1 and M2 as well. On the XWT, the color scale represents the amplitude, the arrows are the phase shift (positive correlation pointing right, anti-correlated pointing left, and the progressive time shift in between), and the bold line delimits the zones with significant correlation (more than two sigma).**



**4.3 Links between ice mass changes and meteorological parameters**

The meteorological parameters have been compared and analyzed with the EOF decomposition of the GRACE ice mass changes, like the climate indices in section 4.2.

In Fig. 4, we plot the decomposed GRACE variations and one meteorological parameter on the same figure and choose the combination of modes that presents the best correlation for each one. This is M1, plus M3 and M4 for TDT (a), and the sum of M1 to M5 for P (b) and M1 to M3 for AI (c). In addition, the corresponding XWT is represented to the right of each plot.

The TDT signal is strongly anti-correlated, -0.88, to the annual mass variations of the combined M1, M3, and M4 modes. At this scale, the thermal availability leads by up to 2 months the ice mass changes. There's also a common oscillation at an 11 years period, whose coefficient is -0.71. (Figure 4a and Table 1)

In Fig. 4b, the combination of the first fifth mode is required to find an interannual correlation between the P and the modes. This correlation, with a coefficient of 0.75, is for an 11 year cycle.

Between 2004 and 2018, the AI is shown to be strongly correlated, with a coefficient of 0.8, to the ice mass changes of the combined modes M1 to M3 with a common periodicity of 7 years. The XWT also shows a strong annual correlation between both signals which has a coefficient of 0.9. (Figure 4c and Table 1)





**Figure 4: Comparison of the cumulated meteorological parameter (in red) with the modes or summation of modes of ice mass changes in Gt (in blue) between 2002 and 2024, and the corresponding XWT analysis. In a) is the TDT compared with M1 + M3 and M4 modes, b) is the P with M1 to M5, and c) is the Al with the sum of M1 to M3. On the XWT, the color scale represents the amplitude, the arrows represent the phase shift (positive correlation pointing right, anti-correlated pointing left, and the progressive time shift in between), and the bold line delimits the zones with significant correlation (more than two sigma).**



**5 Discussion**

Our results show interactions between ice mass variations and variations in climate indices or meteorological parameters.
These interactions are more or less evident depending on the frequency with which we study the phenomenon, and they are
often linked to each other.

Among the three indices, the GBI shows the strongest correlation (-0.85 and -0.91) with mass variations. This is a logical result
given that this index is specifically designed for Greenland. This contrasts with the other two indices, such as the AMO, which
only accounts for a small portion of the waters surrounding Greenland. Similarly, the NAO, although it focuses on atmospheric
pressures like the GBI, is calculated for the entire dipole covering the North Atlantic rather than just Greenland. (Figure 3 and
Table 1)

To understand the complex mechanisms that affect the GIS, we start by examining which meteorological parameters can
intervene on the NAO, GBI, and AMO indices.

The most obvious, temperature, is linked to NAO and AMO. The latter responds to our TDT parameter both for the annual
cycle and for the 11 years period. Surprisingly, the GBI index is only sensitive to temperature for the annual frequency.
However, this index is the most local of the three, and it would seem normal that it is directly influenced by local weather
parameters. Currently, annual temperature variations dominate the signal, such that no longer-term behavior is highlighted.

When we look at precipitation, it is correlated with the NAO and AMO only for periods of 4 years and 11 years. No link
between them is visible with an annual periodicity. Only the GBI index is well anti-correlated to the precipitation fall following
the annual cycle. This relationship is consistent with what was expected since the conditions favorable to precipitation are
when the GBI decreases and vice versa.

The albedo is also sensitive to annual variations of the GBI but with a slight lag of one to two months. This sensitivity is
indirectly related to the atmospheric pressure above Greenland, reflected by the GBI index. The control of atmospheric
pressure, on the amount of precipitation and temperature, positively or negatively influences the reflective quality of the GIS
surface, and with more or less delay, the time for the last precipitation to fall in winter or for the energy available for melting
to cause the surface to darken in summer.

Finally, we would like to know what is the force that generates and maintains these periodic variations. Some are obvious,
such as the annual, others are subtler, such as the period between 8 and 12 years, and finally, others are more complicated to
explain, such as the 4 or 7 year periods and the at least 15 years long period. For this last one, we're aware that it's based on a
very strong hypothesis, that of the observed signal's periodicity. The methodology used was chosen because it allows us to
highlight common periods in our signals. It would take a much longer observation period to ascertain this, however, the results
are realistic and consistent with known climatic phenomena.

We will start by discussing in more detail the two periods for which there are strong and indisputable relationships between
ice mass, meteorological parameters, and climate indices: the 1 year period, and the period between 8 and 12 years. Then, we
will assess the other periods visible in the signal studied, and we will finish by looking at the punctual event detected in M5
of the EOF decomposition.



### 5.1 Annual Cycle

The most evident variability, which is found between almost all of these factors, is a one-year variability. We wondered if it was possible to link the different parameters to propose a synthetic diagram grouping them all and showing their respective participation. By exploring in more detail all the interactions that emerge from our study, we managed to represent a complete

cycle over one year. This annual cycle shows how each of the climatic or meteorological factors intervenes in the variations in ice mass over a year. It is represented in Fig. 5.

To describe how this cycle works, we start at the top (A), at the transition between Summer and Winter. During this period, pressure decreases over Greenland while increasing in the Subtropical High. This pressure gradient strengthens the winds blowing from south to north, allowing humid air from the still-warm surface of the North Atlantic Ocean to move northward.

Consequently, the moisture content in Greenland rises. Ultimately, the combination of reduced pressure from the cyclonic system, the high topography, and the moisture-laden air settling over Greenland leads to expanded cloud cover and a high amount of precipitation.

After one to two months (B), the atmospheric pressure over the Subpolar Low reaches its minimum and at the Subtropical High its maximum. The available energy for melting is also at its minimum. (Kuroda et al., 2022)

Between 1 or 2 months later (C), the albedo reaches its maximum as the fresh white snow cover is at its peak due to the late precipitations and the low thermal availability for melting. In parallel to the albedo and for the same reasons, the peak of mass is reached for the GIS. At this point, the sea surface temperature viewed with AMO is also at its lowest.

Then the cycle inverses itself with increasing atmospheric pressure over Greenland and a weakening of the dipole, encouraged by the low temperature of the ocean which limits evaporation making the air less buoyant.

Three months after the negative AMO's peak (D), precipitations are at their lowest point due to the lack of humidity and the strengthening atmospheric blocking system. It's confirmed by the GBI and NAO indices, 1 or 2 months afterward (E), where the first attains its maximum and the second hits its weakest point of the year. At the same time as the indices, the thermal availability reaches its highest values.

Thus, 1 or 2 months later (F), the albedo decreases because of the lack of humidity, the anticyclonic system over Greenland,

and the high thermal availability. The snow cover is not renewed, the ice darkens and it melts. The GIS mass also attains its lowest value of the year. The mass loss causes are melting, evaporation, sublimation, runoff, and calving. As the AMO index is at its maximum, the warm Atlantic waters exacerbate the calving and reach glaciers' outlet, melting their bottoms, and destabilizing them (Chudley et al., 2023).

The 1 or 2 months lag between the AMO and the temperature or indices can be explained by two phenomena. The heating

imbalance from solar radiation is caused by the difference in latitude and the thermic inertia of the ocean. For a water column, the heat transfer between the atmosphere and the ocean requires 15 days for 10 meters of depth (Zavialov, 2010).



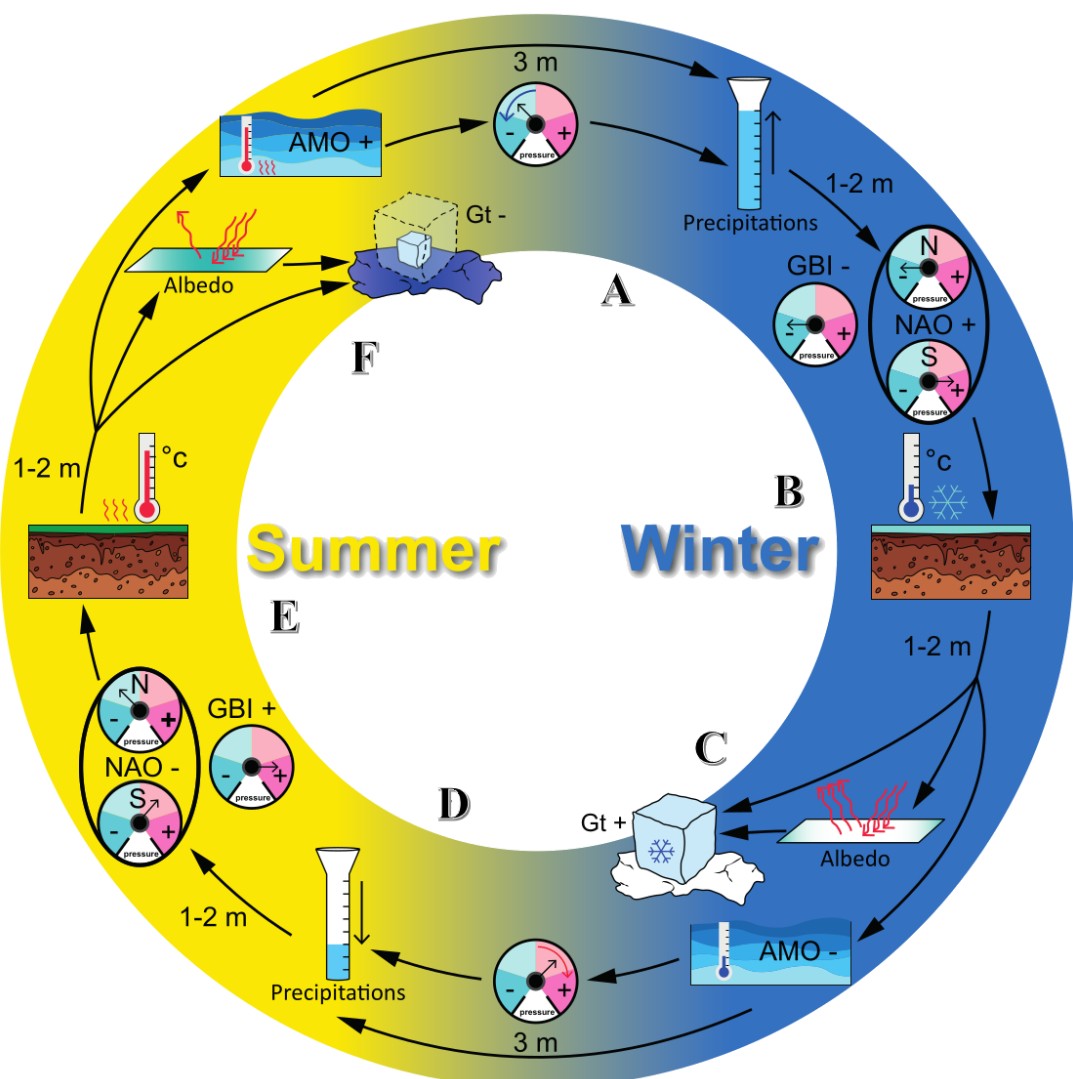

**Figure 5: Summary of the interactions between the GIS, the indices, and the meteorological parameters. This is the annual cycle implied by their respective relationship to each other. The arrows are not direct causal links, they imply that specific events show a positive or negative correlation to one another. A time lag of 1, 2, or 3 months is added to some arrows. This annual cycle is depicted with the summer part of the year on the left and the winter one on the right. The letters A to F help to follow the description in the text.**

## 5.2 Periodicity between 8 and 12 years

The periodicity of 8 and 12 years appears regularly. We investigate a possible link with the well-known solar cycle. Even though the average period of solar activity is 11.2 years, it varies between 8 and 15 years (Ambastha, 2003). It is trivial to think that increases and decreases in solar emissions could influence the ice mass of the ice sheet. We recovered the radio emission from the Sun at a wavelength of 10.7 cm on the Space Weather website of the Government of Canada (Penticton, B.C., Canada, 2024). This dataset is available daily from 2004 to the present.



As shown in Fig. 6, with the example of NAO, a common cycle between 8 and 12 years is visible in the XWT.

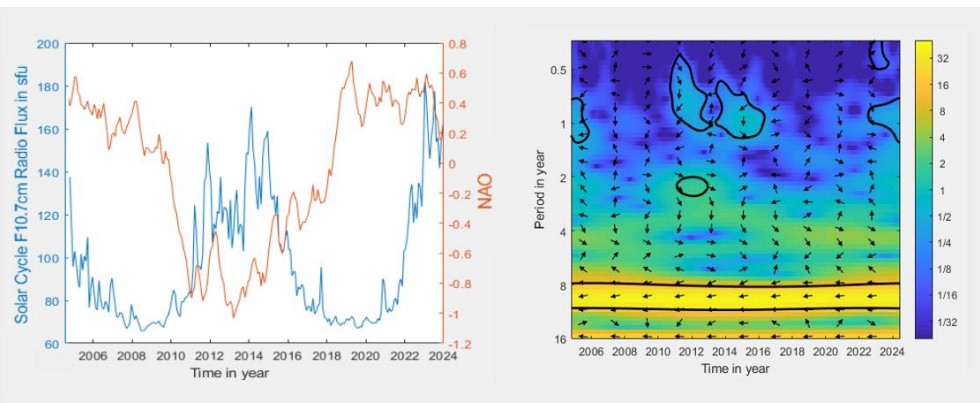

**Figure 6: Comparison of the NAO (in red) and the Solar cycle F10.7cm radio flux (sfu, in blue) between 2002 and 2024. On the XWT, the color scale represents the amplitude, the arrows represent the phase shift (positive correlation pointing right, anti-correlated pointing left, and the progressive time shift in between), and the bold line delimits the zones with significant correlation (more than two sigma).**

When NAO is high the atmospheric pressure dipole will be exacerbated leading to more precipitations in Greenland and so, mass gain on the GIS. It also implies a reduction of the TDT.

Such oscillations around 11 years are visible for almost all parameters and indices we looked for. Thinking of the sun as the driving force of this cycle is therefore a solid track since the sun affects almost all climate actors. However, as this period is half of our observation window, it will be interesting to verify this result when a longer series becomes available. (Drews et al., 2022; Georgieva et al., 2007; Kuroda et al., 2022)

### 5.3 Others periodicities

#### 5.3.1 Periodicity at 4 and 7 years

Interannual cycles, varying from 4 to 7 years, are discernable in the relations between GIS, precipitations, AMO, and albedo.

There's a loss of ice when the GIS has a low albedo or when the AMO index is high. In the first case, the surface is darkening and it absorbs more radiation, so the ice melts and evaporates, or melts and flows away. In parallel, the high sea surface temperature directly melts ice at the outlets of glaciers and increases the calving (Chudley et al., 2023).

There's also a complex link between precipitations and the AMO index. Like in the case of the yearly cycle, we expect the AMO index to lead or at least to be one of the causes of Greenland's precipitations. A large amount of fresh water, here from the precipitations, will slow down the oceanic circulation (Rahmstorf et al., 2015) and then will decrease the oceanic thermic transfer from the south to the north. Thus, heat can be stored in the area where the AMO is calculated. However, even if the behavior is logical, we don't have enough evidence or a simulation to verify this further.

Seeking the cause of these cycles, we found they could be related to atmospheric oscillations already observed for those frequencies (Offermann et al., 2021). Maybe some part comes from a deeper origin. It has been shown that a relation between the angular momentum of the atmosphere with the solid Earth's rotation over a 6 year period exists. It implies the possible link between the dynamical processes of the core, the solid Earth, and the different superficial fluid envelopes (Pfeffer et al., 2023).




### 5.3.2 Periodicity superior to 15 years

During the GRACE and GRACE-FO measurement window, we have shown links between the ice mass variations and some climatic parameters with a periodicity of at least 15 years. This is a very long duration compared to the 22 years of observation. Only the GBI and the AMO indices present a sufficient relationship that is worth being studied.

Remember that the GBI is a blocking index that represents the mean 500 hPa geopotential height over Greenland. It is negatively correlated to the mass variations, with a coefficient of -0.91 for a period superior to 15 years. This strong relation implies the GIS loses mass when the atmospheric pressure is high. This is consistent with the yearly relationship between the two.

The second index, AMO, shows a positive correlation, with a period of at least 15 years, that is not in phase with the GRACE signal. There is a lag of 3.5 years, with a coefficient of correlation of 0.73. This means an ice mass loss over Greenland will influence the AMO signal 3.5 years later. This lag can be related to the time needed to displace the oceanic water from Greenland coasts to the North Atlantic area where the AMO index is calculated. Fresh melted water doesn't mix well with the rest of the ocean, it also weakens the Atlantic Meridional Overturning Circulation (Rahmstorf et al., 2015). Such a delay is not surprising. A similar lag of multiple years between NAO and the sea surface temperature, on the Northeast Continental Shelf, induced by the displacement of water from the Labrador Sea to the rest of the North Atlantic Ocean has already been described (Xu et al., 2015). To confirm the existence of such a long period in the space gravimetric data, we have retrieved 75 years of AMO data. We have computed the cumulated index, and have found, in the FFT analysis of the signal, a peak between 17 and 21 years. Although this does not constitute absolute proof, a periodicity greater than 15 years between the ice mass loss and AMO is plausible.

### 5.4 Non-periodic event

As the two events recovered on M5 are that of mass gain, we can examine the probable link to an extreme precipitation event by an XWT of M5 and the non-cumulated P. (Figure 7)

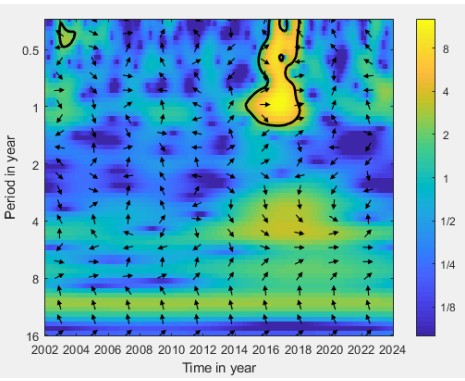

**Figure 7: XWT analysis of the fifth mode and the non-cumulated precipitation signal over Greenland between 2002 and 2024. The color scale represents the amplitude, the arrows represent the phase shift (positive correlation pointing right, anti-correlated pointing left, and the progressive time shift in between), and the bold line delimits the zones with significant correlation (more than two sigma).**



Figure 7 shows a significant positive correlation between the precipitation and the GIS variations in 2017 but not in 2018. This means that the two peaks in M5 can be explained, respectively, by the effect of an extreme weather event or by the gap-filling method linking GRACE and GRACE-FO ice mass variations. Indeed, the second peak is an event taking place during the 11 month gap between the two GRACE(-FO) missions. By using the period preceding the gap, the 2017 event has influenced the Fourier analysis coefficients and, ultimately, the values filling the gap.

The first event, in 2017, is linked to a heavy precipitation event. Specifically, the remnants of Hurricane Nicole in 2016 increased the amount of precipitation over Greenland in winter. Also, in June and August, a storm deposited a snow cover over the ice which in turn increased the albedo and, respectively, delayed and ended the melting season. (Stendel, 2018)

## 6 Conclusion

In this paper, we investigated the temporal variations of the Greenland ice sheet between 2002 and 2023. The declining GIS shows annual and interannual modulations, for which we looked for direct and indirect causes. To do this, we decomposed the ice mass change into five modes (M1 to M5) using an Empirical Orthogonal Function decomposition. The sum of these five modes explains 67.5 % of the total mass variations. We also did a Wavelet Cross-Correlation analysis that allowed us to study the links between the ice mass and climate indices (NOA, GBI, and AMO) and meteorological parameters (Temperature, 435    Precipitation, and Albedo) time series.

The results of this study highlight correlations for several periodicities.

The most obvious is the 1 year periodicity, it is not the simplest to understand. Indeed, each of the parameters considered (ice mass, climate, and weather) presents a link on an annual scale with one or more of the other ones. However, by looking in detail at the different interactions and trying to understand them, we managed to complete an entire cycle over one year, 440    including all the causes of ice mass variations and showing all the links that emerged during the study. We therefore propose a mode of operation for the Ice – Atmosphere – Ocean system in the Greenland region. This cycle is probably a simplified version of the phenomena existing in this region. However, it allows us to understand the main principles of the mechanisms in progress and the sensitivity of the ice cap to the numerous influences coming from the atmosphere and the ocean.

Other periodicities were also highlighted, with values of 4 to 7 years, 11 years, and at least 15 years. The 11 year periodicity can be linked to variations in solar activity. On the other hand, it will be necessary to analyze a longer time series of GRACE solution to ensure the periodicity of at least 15 years is real and to explain its origin.

Finally, our study focused on non-periodic variations. An example is the peak visible in the M5 mode of the GRACE signal 450    decomposition. It shows that the responsiveness of the system to climate change is important, particularly in the case of extreme events or temporary forcing. We can therefore consider non-periodic interannual variations linked to the indices and parameters that we have studied, or linked to other factors that we have not explored.
The indices studied here are calculated near Greenland and their influence in this climatic system is already proven. However, indices from farther away, such as the Pacific Decadal Oscillation or the Southern Oscillation Index (SOI), can also impact the 455    northern atmosphere and hydrosphere. One example of that is the increased likelihood of a positive NAO event when the SOI index is negative and synergizes with a negative AMO (Zhang et al., 2019).

We have examined the temporal aspect of the relations between GIS and its environment, but responses from spatially different parts of Greenland to atmospheric and oceanic forcing should also be studied to better understand where precisely the GIS thickens or is threatened the most by this climatic system.




**Code Availability**

All used codes were in MATLAB language. The code to perform wavelet analysis is the "Cross Wavelet and Wavelet Coherence toolbox", available at https://github.com/grinsted/wavelet-coherence?tab=License-1-ov-file or in another format via http://www.glaciology.net/wavelet-coherence.


**Data Availability**

The GRACE(-FO) COST-G solution is available at the International Centre for Global Earth Models (ICGEM) website: https://icgem.gfz-potsdam.de/sl/temporal.

All three indices, NAO, GBI, and AMO, come from the National Center for Environmental Prediction of the National Oceanic

and Atmospheric Administration (NOAA). For NAO, data and information are found on the Climate Prediction Center and the National Weather Service: https://www.cpc.ncep.noaa.gov/data/teledoc/nao.shtml. The GBI and the AMO indices are available on the Physical Sciences Laboratory (PSL) website, with their respective URL being: https://psl.noaa.gov/gcos_wgsp/Timeseries/GBI_UL/ and https://psl.noaa.gov/data/timeseries/AMO/.

The meteorological parameters, temperature, precipitation, and albedo, are data from the C3S Arctic Regional Reanalysis

(CARRA) and are stored in the Climate Data Store of Copernicus Climate Change Service. https://cds.climate.copernicus.eu/datasets/reanalysis-carra-single-levels?tab=overview

The solar cycle data is available on the Space Weather website of the Government of Canada (Penticton, B.C., Canada, 2024): https://spaceweather.gc.ca/forecast-prevision/solar-solaire/solarflux/sx-5-en.php.

**Author Contribution**

This study has been conceptualized by FC and ML. JD introduced the EOF and Wavelet method whereas FC did the overall methodology. The data curation, the formal analysis, the use of software, and the investigation were done by FC. Figures were created by FC and completed with the ideas of ML. The funding acquisition was done by JD, ML, and FC. FC was supervised by ML, JD, LS, and GR. All the authors have participated in the validation process as well as the writing of the original draft.


**Competing interests**

The authors declare that they have no conflict of interest.

**Acknowledgments**

The authors gratefully thank the Centre National d'Etudes Spatiales (CNES) and the Bureau Gravimétrique International (BGI) for their support. We also thank Etienne Berthier and Anthony Mémin for their fruitful discussions.

**Financial support**



This article's processing charges were covered by the TOSCA project supported by the CNES.


**Review statement**

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
