# Peer review of "Greenland's annual and interannual mass variations from GRACE/GRACE-FO linked with climatic indices"

_EGUsphere, 2025_

## Referee Comment (RC1)

Review of "Greenland's annual and interannual mass variations from GRACE/GRACE-FO linked with climatic indices"

**Summary:** This paper seeks to understand how annual to decadal-scale variability in the mass change of the Greenland Ice Sheet is related to a set of climate indices and parameters. After decomposing the mass change time series into the top five principal components (PCs), numerous cross correlations are performed between the PCs and climate variables using a cross wavelet transform. The most significant relationships with the highest correlations are then discussed. The correlations for annual variability are summarized and some causal relationships investigated in previous studies are noted. I have some major concerns as to the design of the study, the lack of a clear and comprehensive connection to the peer-reviewed literature, and the quality of the writing. Until these major comments are addressed, it is difficult to evaluate many of the specifics of the study.

**Major Comments:**
1.  The methods in this study have the potential to be part of an investigation into causal relationships between Greenland Ice Sheet mass loss and other climate processes (King et al., 2023). This study, however, uses correlations without a comprehensive literature or physics-based discussion of what is already known about these relationships. Though some previous work is mentioned, this paper lacks a clear discussion of the current understanding of the Greenland Ice Sheet in the context of the broader climate system. Such a literature review is important for guiding the reasoning behind which relationships are tested, at which lags, and for which frequencies. Some exploration is appropriate, but calculating a large set of correlations with only the confidence bounds as a guide can easily suggest false or misleading relationships. Such an approach, which is explicitly stated in the paper on lines 210-214, can be referred to as "p-hacking" (e.g., Nuzzo, 2014). This may not have been the intention of the authors; however, this is how the paper reads given that the question, methods, and discussion are not clearly guided by the existing peer-reviewed literature nor a physics-based understanding of the system. Without this guidance, it is unclear which findings are spurious and which may represent real, causal relationships between ice mass loss and climate variables. A hypothesis-driven study based on exploring a knowledge gap in the peer-reviewed literature, such as was done in King et al. (2023), could make for a compelling study, even when similar methods are used. Ideally the paper would add new insight into existing hypotheses or propose new hypotheses. As it is currently written, it is not clear what insight this paper adds that may not already be in the existing literature.

2.  The writing, word choice, and tone need work. The writing lacks conciseness and clarity. The word choices include many casual, non-specific, and sometimes misleading words. This, in part, leads to a tone that is too casual, especially for some sections. See below for an example of how this shows up in a few paragraphs. I hope the example helps illuminate how the writing can be improved throughout the entire manuscript.

This example shows the types of edits I would provide throughout the entire paper. I hope that it is a helpful guide for future editing. In the following, underlines indicate casual wording, *italics indicate vague wording*, **bold indicates confusing or misleading wording**, and [comments are

inserted in square brackets]. This does not include grammar or punctuation issues or fine-combed editing for conciseness and clarity.

Section 5, Lines 280-312

"Our results **show interactions** [correlation is not causation] between ice mass variations and variations in climate indices or meteorological parameters. These **interactions** are more or less evident depending on the frequency with which we study the phenomenon, *and they are often linked to each other*.

Among the three indices, the GBI shows the strongest correlation (-0.85 and -0.91) with mass variations. This is a logical result given that this index is specifically designed for Greenland. This contrasts with the **other two indices, such as the AMO**, [confusing wording] which only accounts for *a small portion* [be more specific] of the waters surrounding Greenland. Similarly, the NAO, although it focuses on atmospheric pressures like the GBI, is calculated for the entire dipole covering the North Atlantic rather than just Greenland. (Figure 3 and Table 1) [Something does not have to be local to have a strong impact. A stronger argument would be based on our physical understanding of the atmosphere and the ice sheet.]

To understand the complex mechanisms that affect the GIS, we start by examining which meteorological parameters *can intervene on* [do you mean "interact with" or "impact"?] the NAO, GBI, and AMO indices. The most obvious, temperature, is **linked** to NAO and AMO. [Linked also implies causation, which has not been shown here.] The latter **responds to** [this again implies causation] our TDT parameter both for the annual cycle and for the 11 years period. Surprisingly, the GBI index is only **sensitive** [also implies causation] to temperature for the annual frequency [cite?]. However, this index is *the most local of the three*, and it would seem normal that it is directly influenced by local weather parameters [cite]. *Currently*, [current to what?] annual temperature variations dominate *the signal*, [which signal?] such that no *longer-term behavior* [what does long-term mean in this context?] is highlighted.

When we look at precipitation, it is correlated with the NAO and AMO only for periods of 4 years and 11 years. No **link** between them is visible with an annual periodicity. Only the GBI index is well anti-correlated to the precipitation **fall** *following* **the annual cycle** [what is "precipitation fall"? What does "following" imply in this context?]. This relationship is consistent with what was expected since the conditions favorable to precipitation are when the GBI decreases and vice versa [cite. Be clear about what are new findings versus what is already in the literature.].

The albedo is also **sensitive** to annual variations of the GBI but with a slight lag of one to two months. This **sensitivity** is indirectly related to the atmospheric pressure above Greenland, reflected by the GBI index [cite if there is literature on this]. **The control of atmospheric pressure, on the amount of precipitation and temperature,** *positively or negatively influences* **the reflective quality of the GIS surface, and with** *more or less delay*, **the time for the last precipitation to fall in winter or for the energy available for melting to cause the surface to darken in summer.** [confusingly worded and needs citations]

Finally, we would like to know what is *the force* that generates and maintains these periodic variations. Some are obvious, such as the annual, others are subtler, such as the period between 8 and 12 years, and finally, others are more complicated to explain, such as the 4 or 7 year periods and the at least 15 years long period. For this last one, we're [contractions are too casual] aware that it's based on a very strong hypothesis [explain and cite], ***that of the observed signal's periodicity***. The methodology used was chosen because it allows us to *highlight* common periods in our signals. It would take a much longer observation period to ascertain *this* [what does "this" refer to?], however, the results are realistic and consistent with known climatic phenomena [explain and cite].

We will start by discussing in more detail the two periods for which there are strong and **indisputable relationships** [significant correlations are not the same as indisputable relationships. More evidence is needed to back up this claim.] between ice mass, meteorological parameters, and climate indices: the 1 year period, and the period between 8 and 12 years. Then, we will **assess** [replace with the word "discuss"? No formal assessment is done.] the other periods visible in the signal studied, and we will finish by looking at the punctual **event** [two events were mentioned earlier] detected in M5 of the EOF decomposition."

**References:**

King, M.A., Lyu, K. & Zhang, X. Climate variability a key driver of recent Antarctic ice-mass change. *Nat. Geosci.* **16**, 1128–1135 (2023). https://doi.org/10.1038/s41561-023-01317-w

Nuzzo, R. Scientific method: Statistical errors. *Nature* **506**, 150–152 (2014). https://doi.org/10.1038/506150a

---

## Author Comment (AC1)

**Reviewer 1 comments,** and our response

Review of "Greenland's annual and interannual mass variations from GRACE/GRACE-FO linked with climatic indices"

Summary: This paper seeks to understand how annual to decadal-scale variability in the mass change of the Greenland Ice Sheet is related to a set of climate indices and parameters. After decomposing the mass change time series into the top five principal components (Pcs), numerous cross correlations are performed between the PCs and climate variables using a cross wavelet transform. The most significant relationships with the highest correlations are then discussed. The correlations for annual variability are summarized and some causal relationships investigated in previous studies are noted. I have some major concerns as to the design of the study, the lack of a clear and comprehensive connection to the peer-reviewed literature, and the quality of the writing. Until these major comments are addressed, it is difficult to evaluate many of the specifics of the study.

We would like to thank reviewer 1 for his constructive comments, pointing out the lack of publications supporting the effects and causes mentioned in our manuscript and derived directly from the observed co-relationships. Some parts have been rewritten to make the text more concise and understandable.

**Major Comments:**

**1. The methods in this study have the potential to be part of an investigation into causal relationships between Greenland Ice Sheet mass loss and other climate processes (King et al., 2023).**

Thank you for your detailed and pertinent comment. You raise some important points about the methodology and contribution of this study. Here are a few points to address your concerns:

1) **Lack of literature review and physical basis:**

You are right to stress the importance of a thorough literature review and physics-based discussion to contextualize the relationships studied. This would help to better justify methodological choices, such as the timeframes and frequencies analyzed, and avoid misinterpretations. We recognize that this shortcoming may make the results less convincing, and pledge to incorporate more climatic literature into the state of the art, as well as in the discussion **of** this new version of the manuscript.

You will find, following this, part of the added references and their contribution. If we have missed relevant references, please do not hesitate to enlighten us.

(King and Christoffersen, 2024): Use the same approach of cumulative indices, but for Antarctica, it is the study that follows the one from King et al., 2023 (reference on which our study's methodology was built upon).

(Lean, 2017): Provides an explanation of the principles behind the dynamical response of the atmosphere with an increase in solar irradiance. The proposed explanation shows that an increase of 0.1% of solar irradiance can increase Earth's global surface temperature by around 0.1°C. It is also indicated that the atmospheric responses are stronger 0.3°C at 20 km, and that equator–to–pole, as well as the vertical thermal gradient, are altered, modifying the dynamical processes of the different climatic systems.

(Hanna et al., 2013): Correlate the GBI, NAO, and AMO indices to Greenland's runoff and temperature. Showing that GBI has a higher correlation coefficient than NAO.

(Preece et al., 2022): Show there exist three blocking patterns that produce increasing melt in different locations depending on the type of blocking.

(Hanna et al., 2015): Indicate that NAO tends to decrease in summer and has more variability in winter. They show it is happening while the GBI tends to increase in summer and be more variable in winter.

(Hanna et al., 2018): In this communication, they show that the recent increase in blocking events in summer may be influenced by the positive AMO, which is related to negative NAO. (Davini et al., 2012)

(Lewis et al., 2021): Observe that strong blocking increases precipitation grain sizes, which in turn reduces the albedo. This also causes an increase in surface temperature, fewer storms, more shortwave radiation, and maximizes the albedo feedback. (Box et al., 2012; Dozier et al., 1981; Tedesco et al., 2011)

(Silva et al., 2022): Use GBI and NAO altogether. They show that a variable surface warming is happening in winter, and that there is an increase in shortwave radiation when higher pressure stands over Greenland.

(Tedesco et al., 2013): Indicate that persistent anticyclonic conditions, NAO's anomalies, and surface albedo and temperatures were drivers of the massive ice loss event of 2012.

(Fettweis et al., 2008): show that Greenland's atmospheric variability and ice sheet melting are correlated to NAO and that it explains extreme temperature in 2003 and melting in 2007.

(Shang et al., 2022): Found that summer mass variations are correlated to the NAO, with temperature-associated precipitation and runoff.

(Li et al., 2022): use the EOF method to find a correlation between ice mass variation and indices such as NAO and AMO. They suggest a relation between the Pacific Decadal Oscillation and the Icelandic Low. The NAO seems to be linked to the West-East dipole of precipitation (correlation in the west and anti-correlation in the east), and AMO to the runoff and temperature.

(Bjørk et al., 2018): have the same results for NAO.

(Sun et al., 2019): observe the close annual and interannual relationship between NAO, the sea surface temperature (SST), which is reflected by the AMO, and the Atlantic Meridional Overturning Circulation.

(Chen et al., 2015): also add the importance of El Niño–Southern Oscillation index to the relation between NAO and SST.

(Kim et al., 2021): show that the AMO is interconnected to the AMOC with a periodicity of 10 to 30 years. It was observed that a positive AMOC induces, with a delay of 7 years, a positive AMO thanks

to meridional heat transport. When the AMO is at his highest, it leads to a reduced water density in the Atlantic sinking area and is the origin of the negative phase of the AMOC.

(Jo et al., 2024): indicate that SST are projected to intensify in summer due to increased heat input and less mixing, whereas in winter they will diminish with the strengthening of vertical mixing.

(Noël et al., 2014): The effect of SST on Greenland remains coastal. They do not appear to influence the central part of the GIS.

(Wood et al., 2021): The SST, salinity, and the geometry of glaciers' ocean-terminating outlets are the main parameters behind their calving behavior.

(Carnahan et al., 2022): observe that ocean forcing is primordial, but not necessary for future mass loss.

(Carrivick et al., 2023): mention the evolution of glaciers. After their retreat, ocean-terminating glaciers transform into land-terminating glaciers, which diminish the discharge term in the mass balance and, in turn, slow the ice loss. However, a land-terminated glacier can change to become lake-terminated, which creates strong calving and accelerates the mass loss.

References:

Bjørk, A. A., Aagaard, S., Lütt, A., Khan, S. A., Box, J. E., Kjeldsen, K. K., Larsen, N. K., Korsgaard, N. J., Cappelen, J., Colgan, W. T., Machguth, H., Andresen, C. S., Peings, Y., and Kjær, K. H.: Changes in Greenland's peripheral glaciers linked to the North Atlantic Oscillation, Nature Clim Change, 8, 48–52, https://doi.org/10.1038/s41558-017-0029-1, 2018.

Box, J. E., Fettweis, X., Stroeve, J. C., Tedesco, M., Hall, D. K., and Steffen, K.: Greenland ice sheet albedo feedback: thermodynamics and atmospheric drivers, The Cryosphere, 6, 821–839, https://doi.org/10.5194/tc-6-821-2012, 2012.

Carnahan, E., Catania, G., and Bartholomaus, T. C.: Observed mechanism for sustained glacier retreat and acceleration in response to ocean warming around Greenland, The Cryosphere, 16, 4305–4317, https://doi.org/10.5194/tc-16-4305-2022, 2022.

Carrivick, J. L., Boston, C. M., Sutherland, J. L., Pearce, D., Armstrong, H., Bjørk, A., Kjeldsen, K. K., Abermann, J., Oien, R. P., Grimes, M., James, W. H. M., and Smith, M. W.: Mass Loss of Glaciers and Ice Caps Across Greenland Since the Little Ice Age, Geophysical Research Letters, 50, https://doi.org/10.1029/2023gl103950, 2023.

Chen, S., Wu, R., and Chen, W.: The Changing Relationship between Interannual Variations of the North Atlantic Oscillation and Northern Tropical Atlantic SST, Journal of Climate, 28, 485–504, https://doi.org/10.1175/jcli-d-14-00422.1, 2015.

Davini, P., Cagnazzo, C., Neale, R., and Tribbia, J.: Coupling between Greenland blocking and the North Atlantic Oscillation pattern, Geophysical Research Letters, 39, https://doi.org/10.1029/2012gl052315, 2012.

Dozier, J., Schneider, S. R., and McGinnis, D. F.: Effect of grain size and snowpack water equivalence on visible and near-infrared satellite observations of snow, Water Resources Research, 17, 1213–1221, https://doi.org/10.1029/wr017i004p01213, 1981.

Fettweis, X., Mabille, G., and Erpicum, M.: Circulations atmosphériques et anomalies de fonte à la surface de la calotte glaciaire du Groenland, Bulletin de la Société Géographique de Liège, 51, 2008.

Hanna, E., Jones, J. M., Cappelen, J., Mernild, S. H., Wood, L., Steffen, K., and Huybrechts, P.: The influence of North Atlantic atmospheric and oceanic forcing effects on 1900–2010 Greenland summer climate and ice melt/runoff, Intl Journal of Climatology, 33, 862–880, https://doi.org/10.1002/joc.3475, 2013.

Hanna, E., Cropper, T. E., Jones, P. D., Scaife, A. A., and Allan, R.: Recent seasonal asymmetric changes in the NAO (a marked summer decline and increased winter variability) and associated changes in the AO and Greenland Blocking Index, Int. J. Climatol, 35, 2540–2554, https://doi.org/10.1002/joc.4157, 2015.

Hanna, E., Fettweis, X., and Hall, R. J.: Brief communication: Recent changes in summer Greenland blocking captured by none of the CMIP5 models, The Cryosphere, 12, 3287–3292, https://doi.org/10.5194/tc-12-3287-2018, 2018.

Jo, A. R., Lee, J., Sharma, S., and Lee, S.: Season-Dependent Atmosphere-Ocean Coupled Processes Driving SST Seasonality Changes in a Warmer Climate, Geophysical Research Letters, 51, https://doi.org/10.1029/2023gl106953, 2024.

Kim, H., An, S., and Kim, D.: Timescale-dependent AMOC–AMO relationship in an earth system model of intermediate complexity, Intl Journal of Climatology, 41, https://doi.org/10.1002/joc.6926, 2021.

King, M. A. and Christoffersen, P.: Major Modes of Climate Variability Dominate Nonlinear Antarctic Ice-Sheet Elevation Changes 2002–2020, Geophysical Research Letters, 51, https://doi.org/10.1029/2024gl108844, 2024.

Lean, J. L.: Sun-Climate Connections, in: Oxford Research Encyclopedia of Climate Science, Oxford University Press, https://doi.org/10.1093/acrefore/9780190228620.013.9, 2017.

Lewis, G., Osterberg, E., Hawley, R., Marshall, H. P., Meehan, T., Graeter, K., McCarthy, F., Overly, T., Thundercloud, Z., Ferris, D., Koffman, B. G., and Dibb, J.: Atmospheric Blocking Drives Recent Albedo Change Across the Western Greenland Ice Sheet Percolation Zone, Geophysical Research Letters, 48, https://doi.org/10.1029/2021gl092814, 2021.

Li, Z., Chao, B. F., Zhang, Z., Jiang, L., and Wang, H.: Greenland Interannual Ice Mass Variations Detected by GRACE Time-Variable Gravity, Geophysical Research Letters, 49, https://doi.org/10.1029/2022GL100551, 2022.

Noël, B., Fettweis, X., Van De Berg, W. J., Van Den Broeke, M. R., and Erpicum, M.: Sensitivity of Greenland Ice Sheet surface mass balance to perturbations in sea surface temperature and sea ice cover: a study with the regional climate model MAR, The Cryosphere, 8, 1871–1883, https://doi.org/10.5194/tc-8-1871-2014, 2014.

Preece, J. R., Wachowicz, L. J., Mote, T. L., Tedesco, M., and Fettweis, X.: Summer Greenland Blocking Diversity and Its Impact on the Surface Mass Balance of the Greenland Ice Sheet, JGR Atmospheres, 127, https://doi.org/10.1029/2021jd035489, 2022.

Shang, P., Su, X., and Luo, Z.: Characteristics of the Greenland Ice Sheet Mass Variations Revealed by GRACE/GRACE Follow-On Gravimetry, Remote Sensing, 14, 4442, https://doi.org/10.3390/rs14184442, 2022.

Silva, T., Abermann, J., Noël, B., Shahi, S., Van De Berg, W. J., and Schöner, W.: The impact of climate oscillations on the surface energy budget over the Greenland Ice Sheet in a changing climate, The Cryosphere, 16, 3375–3391, https://doi.org/10.5194/tc-16-3375-2022, 2022.

Sun, C., Li, J., Kucharski, F., Xue, J., and Li, X.: Contrasting spatial structures of Atlantic Multidecadal Oscillation between observations and slab ocean model simulations, Clim Dyn, 52, 1395–1411, https://doi.org/10.1007/s00382-018-4201-8, 2019.

Tedesco, M., Fettweis, X., Van Den Broeke, M. R., Van De Wal, R. S. W., Smeets, C. J. P. P., Van De Berg, W. J., Serreze, M. C., and Box, J. E.: The role of albedo and accumulation in the 2010 melting record in Greenland, Environ. Res. Lett., 6, 014005, https://doi.org/10.1088/1748-9326/6/1/014005, 2011.

Tedesco, M., Fettweis, X., Mote, T., Wahr, J., Alexander, P., Box, J. E., and Wouters, B.: Evidence and analysis of 2012 Greenland records from spaceborne observations, a regional climate model and reanalysis data, The Cryosphere, 7, 615–630, https://doi.org/10.5194/tc-7-615-2013, 2013.

Wood, M., Rignot, E., Fenty, I., An, L., Bjørk, A., Van Den Broeke, M., Cai, C., Kane, E., Menemenlis, D., Millan, R., Morlighem, M., Mouginot, J., Noël, B., Scheuchl, B., Velicogna, I., Willis, J. K., and Zhang, H.: Ocean forcing drives glacier retreat in Greenland, Sci. Adv., 7, https://doi.org/10.1126/sciadv.aba7282, 2021.

**2) Risk of "p-hacking":**

We understand your concerns about the exploratory approach and the risk of suggesting false or misleading relationships. While it was not our intention to practice "p-hacking", we recognize that the current methodology could give this impression. We plan to reformulate our approach so that it is guided more by clear hypotheses and existing knowledge, while limiting the number of correlations tested to avoid abusive interpretations.

A typical example is the observations made for the 'Periodicity superior to 15 years' which, from our point of view, is simply a period to watch over the next few decades and cannot be confirmed in the context of this time series, which is far too short to go any further. Aware of the confusion that these remarks may cause, we have removed this paragraph '5.3.2 Periodicity superior to 15 years' from the final text.

**3) Contribution to existing literature:**

We appreciate your suggestion to structure the study around a gap in the literature, as in King et al. (2023). This would clarify the added value of our work. We will revise our introduction and discussion to better highlight the new perspectives or hypotheses this study proposes, in relation to recent research on mass loss from the Greenland ice sheet, the climatic indices and meteorological parameters.

**4) Writing clarity and precision:**

We recognize that we lacked conciseness, clarity and that our word choice could have been misleading. We will rewrite the manuscript to take into account your edits and more.

In summary, we take your comments very seriously and are committed to improving the manuscript by strengthening its anchorage in the existing literature, clarifying its objectives and refining the methodology to avoid any ambiguity. We thank you again for your constructive suggestions, which will help to reinforce the quality and rigor of our work.

---

## Author Comment (AC2)

**Reviewer 2 comments,** and our response

This manuscript presents a detailed analysis of Greenland Ice Sheet (GIS) mass variations using GRACE/GRACE-FO data (2002–2023) and investigates their connections with climatic indices (NAO, GBI, AMO) and meteorological parameters (temperature, precipitation, albedo). The study employs Empirical Orthogonal Function (EOF) decomposition and wavelet analysis to identify dominant modes of variability and their relationships with external forcings. The topic is timely and relevant to understanding GIS mass balance under climate change. While the methodology is generally sound, some aspects require clarification, validation, and refinement to strengthen the conclusions.

The use of GRACE/GRACE-FO data (COST-G solutions) is appropriate, and the handling of gaps (interpolation and FFT-based gap-filling) is reasonable. Inclusion of multiple climatic indices (NAO, GBI, AMO) and meteorological parameters (temperature, precipitation, albedo) provides a holistic view of GIS mass balance drivers. The 3.5-year lag between AMO and GIS mass loss is an interesting result, plausibly explained by freshwater transport timescales. The proposed annual cycle (Fig. 5) synthesizes interactions between atmospheric, oceanic, and cryospheric processes coherently.

There are still some major issues that should be properly addressed before consideration of publication.

We would like to thank reviewer 2 for his constructive remarks on the lack of clarity and precision of the different parts of our study. Parts of the manuscript have been rewritten to take into account those comments.

- **The higher EOF modes (M2–M5) are less interpretable, and their physical significance is unclear. M5's peaks in 2017–2018 could reflect interpolation artifacts rather than real signals.**
  - **M2-M5:**

    It is true that our explanation was insufficient in regard to why we chose to use EOF modes (M2-M5). We will rewrite the manuscript to add a more thorough explanation.

    Here we explain the points that affected our reasoning and allowed us to confirm which mode mainly shows a real signal.

    We simulate the null hypothesis of temporally uncorrelated structure through randomization of the temporal dimension. We observed that the eigenvalue from the first 12 modes exceeds that of the one generated through shuffling. We also verified if the modes were different from autocorrelated random processes by comparing the frequency content to that of red noise. Modes M1 to M5 had stronger frequencies than the red noise. We confirm that the modes are distinct from one another through North's test. In that test, Figure A, only the first four modes are shown to be distinct from one another, and their error bars do not overlap with the following mode. We use this last conservative test as a basis to use M1 to M4, and added M5, which is at the limit, passing the first two tests and showing a different behavior from other modes with its two peaks.

[Figure]

Figure A: Eigenvalues of the first ten EOF modes with their error bars using North's test. All modes above the green line are distinct from one another (M1 to M4), the red line shows the overlap between M5 and M6, and all subsequent modes are also overlapping with each other.

It also seems possible to link the EOF modes of mass variations to the EOF modes of Temperature and Precipitation variations. M2 from the EOF of Greenland mass changes is somewhat correlated to mode 2 of the EOF of Greenland's precipitation, and M3 and M4 of mass changes to mode 1 of Greenland's temperature (TDT). This confirms what was shown in Table 1 and Figure 3. (Figure B)

[Figure]

Figure B: The EOF mode of the ice sheet mass changes is shown in blue, and the EOF mode of the meteorological parameters (Precipitation and Temperature) is shown in red. In A), mass change mode M2 is compared to the mode 2 of Precipitation, and in B) is the comparison of the M3 and M4 of mass changes with the mode 1 of Temperature.

- o **M5's peaks:**

We lacked precision on the dates of the two peaks; we will address this issue. The first one is in November 2016, and the second one is in November 2017, during the 11-month gap (July 2017 - May 2018) between the two satellite missions.

The two peaks with 50 and 20 Gt each are stronger than the high-frequency noise-like signal of an amplitude of ~10 Gt.

We agree that the second peak is an interpolation artifact. The peak disappears when we alter the chosen preceding year for the interpolation (e.g., using 2015-2016 instead of 2016-2017).

Nonetheless, the first peak is not in this gap or any other lacking month (e.g., September-October 2016). This peak also shows high correlation with precipitation (between 2016 and 2017) in the Cross-Correlation Wavelet analysis (Figure 7). Moreover, we know that in October 2016, a large amount of precipitation was recorded over Greenland (Stendel, 2018). It was in that period that the remnants of Hurricane Nicole reached Greenland's south-eastern coast (Stendel, 2018). Even with the October month lacking, the effect of those events seems to linger till November on the gravimetric data.

Stendel, M.: Polar Portal Season Report 2017, DMI, GEUS, DTU-Space and DTU-Byg, 2018.

- **The 11-year solar cycle link seems plausible but speculative given the short (22-year) dataset**

  Albeit short, 22 years of data, our results are in line with what several studies (line 370-373 "Drews et al., 2022; Georgieva et al., 2007; Kuroda et al., 2022", or Mares et al., (2022) have already showcased for the correlations between solar cycles and indices such as NAO and AMO. Judith Lean (2017) provides an explanation of the principles behind the dynamical response of the atmosphere with an increase in solar irradiance. The proposed explanation shows that an increase of 0.1% of solar irradiance can increase Earth's global surface temperature by around 0.1°C. It is also indicated that the atmospheric responses is stronger 0.3°C at 20 km, and that equator–to–pole, as well as the vertical thermal gradient, are altered, modifying the dynamical processes of the different climatic systems. We will add these last two references to the article.

  Drews, A., Huo, W., Matthes, K., Kodera, K., and Kruschke, T.: The Sun's role in decadal climate predictability in the North Atlantic, Atmos. Chem. Phys., 22, 7893–7904, https://doi.org/10.5194/acp-22-7893-2022, 2022.

  Georgieva, K., Kirov, B., Tonev, P., Guineva, V., and Atanasov, D.: Long-term variations in the correlation between NAO and solar activity: The importance of north–south solar activity asymmetry for atmospheric circulation, Advances in Space Research, 40, 1152–1166, https://doi.org/10.1016/j.asr.2007.02.091, 2007.

  Lean, J.: Sun-Climate Connections, Oxford Research Encyclopedia of Climate Science., https://doi.org/10.1093/acrefore/9780190228620.013.9, 2017.

  Kuroda, Y., Kodera, K., Yoshida, K., Yukimoto, S., and Gray, L.: Influence of the Solar Cycle on the North Atlantic Oscillation, JGR Atmospheres, 127, e2021JD035519, https://doi.org/10.1029/2021JD035519, 2022.

  Mares, C., Dobrica, V., Mares, I., Demetrescu, C.: Solar Signature in Climate Indices, *Atmosphere*, *13*, 1898, https://doi.org/10.3390/atmos13111898, 2022.

- **The >15-year signal is intriguing but statistically uncertain**

    We will remove mentions of this intriguing >15-year signal.

- **The cumulative approach for indices (NAO, GBI, AMO) is innovative but lacks a clear physical basis. How do cumulative indices better represent mass balance than raw anomalies?**

    This approach allows the observation of a time-integrated variable, which is the cumulated effect of an index, and not each instantaneous event separated from one another, as in the raw index. When they are linked to one another, a raw index event is related to the instantaneous speed of mass variations. Thus, the time-integrated indices are related to the observed time-integrated mass variations from gravimetric satellites, rather than the mass flux as the raw indices do.

    This approach has already been used by Mazzarella A., 2013, between the NAO and sea surface temperature, or also by King M. and Christoffersen P., 2024, for ENSO index, GRACE data, as well as altimetry data. In Line 90-93 we explain this approach was also used in the case of Antarctica (Kim et al., 2020; King et al., 2023; Paolo et al., 2018).

    Mazzarella, A.: Time-integrated North Atlantic Oscillation as a proxy for climatic change, *Natural Science*, 5, 149-155, https://doi.org/10.4236/ns.2013.51A023, 2013.

    Kim, B.-H., Seo, K.-W., Eom, J., Chen, J., and Wilson, C. R.: Antarctic ice mass variations from 1979 to 2017 driven by anomalous precipitation accumulation, Sci Rep, 10, 20366, https://doi.org/10.1038/s41598-020-77403-5, 2020.

    King, M. A., Lyu, K., and Zhang, X.: Climate variability a key driver of recent Antarctic ice-mass change, Nat. Geosci., 16, 1128–1135, https://doi.org/10.1038/s41561-023-01317-w, 2023.

    King, M. A., & Christoffersen, P.: Major modes of climate variability dominate nonlinear Antarctic ice-sheet elevation changes 2002–2020, *Geophysical Research Letters*, 51, e2024GL108844, https://doi.org/10.1029/2024GL108844, 2024.

    Paolo, F. S., Padman, L., Fricker, H. A., Adusumilli, S., Howard, S., and Siegfried, M. R.: Response of Pacific-sector Antarctic ice shelves to the El Niño/Southern Oscillation, Nature Geosci, 11, 121–126, https://doi.org/10.1038/s41561-017-0033-0, 2018.

- **No independent validation (e.g., altimetry, regional climate models) is provided to cross-check GRACE-derived mass changes.**

    We will add a paragraph to show that our results are in accordance with what has been found in previous studies, such as that of altimetry data from Khan et al. (2024) and the results from the IMBIE team of 2023.

    You can find in Figure C, our GRACE-derived mass variations, in red, the altimetry one from Khan et al., 2024, in green, and the one from the mix of methods (Altimetry, Gravimetry, Input-Output method) of the IMBIE team, 2023, in blue. We also show the linearly detrended version of our data against that of Khan et al. (2024) in Figure D.

[Figure]

Figure C: Greenland mass variations in Gt since 1992 to 2024. This study's GRACE-derived mass variations are in red; Khan et al. (2024) altimetry data are in green, and the IMBIE team results from 2023 are in blue.

[Figure]

Figure D: Greenland linearly detrended mass variations, in red are the results of this study, and in green the ones from the altimetry study of Khan et al., 2024.

Khan, S. A., Seroussi, H., Morlighem, M., Colgan, W., Helm, V., Cheng, G., Berg, D., Barletta, V. R., Larsen, N. K., Kochtitzky, W., van den Broeke, M., Kjær, K. H., Aschwanden, A., Noël, B., Box, J. E., MacGregor, J. A., Fausto, R. S., Mankoff, K. D., Howat, I. M., Oniszk, K., Fahrner, D., Løkkegaard, A., Lippert, E. Y. H., and Hassan, J.: Smoothed monthly Greenland ice sheet elevation changes during 2003–2023, Earth Syst. Sci. Data Discuss. [preprint], https://doi.org/10.5194/essd-2024-348, in review, 2024.

Otosaka, I. N., Shepherd, A., Ivins, E. R., Schlegel, N.-J., Amory, C., van den Broeke, M., Horwath, M., Joughin, I., King, M., Krinner, G., Nowicki, S., Payne, T., Rignot, E., Scambos, T., Simon, K.

M., Smith, B., Sandberg Sørensen, L., Velicogna, I., Whitehouse, P., A, G., Agosta, C., Ahlstrøm, A. P., Blazquez, A., Colgan, W., Engdahl, M., Fettweis, X., Forsberg, R., Gallée, H., Gardner, A., Gilbert, L., Gourmelen, N., Groh, A., Gunter, B. C., Harig, C., Helm, V., Khan, S. A., Konrad, H., Langen, P., Lecavalier, B., Liang, C.-C., Loomis, B., McMillan, M., Melini, D., Mernild, S. H., Mottram, R., Mouginot, J., Nilsson, J., Noël, B., Pattle, M. E., Peltier, W. R., Pie, N., Sasgen, I., Save, H., Seo, K.-W., Scheuchl, B., Schrama, E., Schröder, L., Simonsen, S. B., Slater, T., Spada, G., Sutterley, T., Vishwakarma, B. D., van Wessem, J. M., Wiese, D., van der Wal, W., and Wouters, B.: Mass Balance of the Greenland and Antarctic Ice Sheets from 1992 to 2020, ESSD – Ice/Glaciology, https://doi.org/10.5194/essd-2022-261, 2023.

- **Some methodological details are unclear (e.g., "time-weighted thermal availability," wavelet significance thresholds)."**

    We will make things clearer in the manuscript, but here is an explanation.

    The "time-weighted thermal availability" is our transformed Temperature parameter, TDT, a time-integrated temperature parameter. It corresponds to the mean of all positive temperatures ($T > 0°c$) of a month (i), multiplied by the time duration where $T > 0°c$ is reached during this same month (i) (Line 133-138 explains this as Eq. 1).

    On the Cross-Wavelet analysis, the black contours are estimated, using Monte Carlo simulation, against red noise, with a 2-sigma threshold. That translates to a 5% significance level, meaning there is an only a 5% chance that the circled areas happened by chance, and a 95% confidence that all other zones beyond the black contours are noise. (Grinsted et al., 2004)

Overall, this study makes a valuable contribution to understanding GIS mass variability and its climatic drivers. The methodology looks sound, and the results seems plausible, but some claims (e.g., solar cycle link, >15-year periodicity) require caution due to dataset limitations.